# Fast and Accurate $k$-means++ via Rejection Sampling

**Vincent Cohen-Addad**[*]
Google Research
cohenaddad@google.com

**Silvio Lattanzi**[*]
Google Research
silviol@google.com

**Ashkan Norouzi-Fard**[*]
Google Research
ashkannorouzi@google.com

**Christian Sohler**[*†]
University of Cologne
csohler@uni-koeln.de

**Ola Svensson**[*]
EPFL
ola.svensson@epfl.ch

## Abstract

$k$-means++ [3] is a widely used clustering algorithm that is easy to implement, has nice theoretical guarantees and strong empirical performance. Despite its wide adoption, $k$-means++ sometimes suffers from being slow on large data-sets so a natural question has been to obtain more efficient algorithms with similar guarantees. In this paper, we present a near linear time algorithm for $k$-means++ seeding. Interestingly our algorithm obtains the same theoretical guarantees as $k$-means++ and significantly improves earlier results on fast $k$-means++ seeding. Moreover, we show empirically that our algorithm is significantly faster than $k$-means++ and obtains solutions of equivalent quality.

## 1 Introduction

Clustering is a fundamental task in machine learning with many applications in data analysis and machine learning. One particularly important variant is $k$-means clustering: Given a set of $n$ points in $\mathbb{R}^d$ the goal is to find a partition of the points into $k$ clusters such that the sum of squared distance to the cluster centers (which are the means of the clusters) is minimized.

A popular method to compute a good clustering with respect to the $k$-means objective is the K-MEANS++ algorithm [3]. The algorithm is a combination of a randomized procedure for finding a set of $k$ starting centers (often referred to as the *seeding*) with the classic local improvement algorithm by Lloyd [20]. The seeding step samples the first center uniformly at random. In the remaining iterations the algorithm samples a center from the $D^2$-distribution, where the probability of sampling a point is proportional to the squared distance to the current set of centers.

The K-MEANS++ algorithm is easy to implement, has strong theoretical guarantees (an $O(\log k)$ approximation guarantee), and performs empirically well. However, the running time of $\Theta(dnk)$[3] becomes impractical for many very large datasets. Therefore, a lot of previous work focused on speeding up the K-MEANS++ seeding [4, 5] as well as Lloyd's algorithm [14, 10, 22, 13].

To obtain a fast seeding algorithm, Bachem et al. [4] and [5] use an MCMC algorithm to generate a set of $k$ centers that follows the K-MEANS++ distribution. They provide different versions of their algorithm that provide trade-offs between theoretical guarantees and empirical running time. Interestingly, under certain assumptions on the inputs and when $k$ is small, their algorithm may even run in sublinear time in the input size. However, all versions of their algorithms have a

---

[*]Equal contribution

[†]Work was partially done while author was visiting researcher at Google Research, Switzerland.

[3]Even assuming a constant number of Lloyd's algorithm steps.

running time of $\Omega(k^2)$[4] and so it does not scale for massive datasets and moderate values of $k$ (i.e. $500 - 1000$). Another important drawback from their results is that the guarantees on the quality of the solution output by their algorithms are weaker than the original K-MEANS++ guarantee (since their approximation is additive in the worst case).

**Our contribution** In this paper we present a new algorithm that provably achieves near-linear running time while offering similar guarantees as the original K-MEANS++ algorithm. In particular:

- We introduce a new seeding algorithm that for constant $\varepsilon > 0$ has near-linear running time $\widetilde{O}\left(nd + (n\log(\Delta))^{1+\varepsilon}\right)$ and returns a $O_\varepsilon(\log k)$ approximate solution, where $n$ is the number of points in the dataset and $\Delta$ is the ratio between the maximum distance and the minimum distance between two points in the dataset, see Corollary 5.5. Our algorithm also has the advantage that, in the stated running time, it computes the solution for *all* values of $k = 1, 2, \ldots, n$.
- We compare the performances of our seeding technique with the baselines K-MEANS++ and AFKMC2 from [4] on various classic datasets. Our algorithms outperform the baselines even for moderate values of $k$ (e.g.: $k = 1000$) and the speed-up is an order of magnitude for larger values of $k$[5]. In addition, our algorithms output solutions of similar costs as K-MEANS++ (as our theoretical results predict).

The main idea behind our method is to use an embedding into a collection of trees to approximate the distances between the input points, and then leverage the tree structure to speed-up the $D^2$-sampling of K-MEANS++. To ensure that our sampling, which uses the approximate tree distances, leads to a solution that is competitive with respect to K-MEANS++ on the *original* data, we "emulate" the $D^2$-distribution on the original data by additionally using rejection sampling. More concretely,
– We first develop a new seeding algorithm FASTK-MEANS++ that computes in $\widetilde{O}(nd)$ time a solution. The near linear running time is obtained by first approximating the squared Euclidean distance using a multi-tree embedding and then by showing that one can efficiently perform $D^2$-sampling with respect to multi-tree distances.
– We then argue that one can use our sampling technique on multi-tree distances in combination with rejection sampling so as to reproduce the same distribution as used by K-MEANS++ on the original distances and so to achieve the same approximation guarantees that K-MEANS++. To ensure a fast running time, we calculate the rejection probability by using locality-sensitive hashing (LSH) to approximately determine the nearest neighbor w.r.t. the original distances.
– Finally, we show that our LSH based rejection sampling algorithm computes a solution with the same expected approximation guarantee of $O(\log k)$ as the basic K-MEANS++ algorithm.

## 2 Preliminaries

**Basic notation.** We denote by $P \subseteq \mathbb{R}^d$ the set of $n$ input points in a $d$ dimensional space and let $\Delta$ be the ratio between the maximum distance and the minimum distance between two point in the dataset. The Euclidean distance between two points $x, y \in \mathbb{R}^d$ is denoted by $\text{DIST}(x, y) = ||x - y||_2$. We also let $\text{DIST}(x, C) = \min_{y \in C} \text{DIST}(x, y)$ be the distance of $x$ to the closest point in $C$. The goal in the $k$-means problem is to choose a set of $k$ centers $C \subseteq \mathbb{R}^d$ minimizing $\sum_{x \in P} \text{DIST}(x, C)^2$.

**Tree embeddings.** Tree embedding is a well-known technique used in many different clustering problems (see for example [6]). We now explain a simple version that is similar to [18] that will be used in our algorithm. We first compute an upper bound MAXDIST on the maximum distance between two points within a factor of 2.[6] Second, we add a random shift $0 \le s \le$ MAXDIST to each coordinate of all input points[7]. Let $x \in P$ be any point in the data set. The root of the tree (at

height zero) represents an axis-aligned cube of side length 2MAXDIST centered at $x$. By selection, note that all the input points are inside this cube and we say that they *belong to* this node of the tree. We then partition this cube into $2^d$ axis-aligned subcubes of side length MAXDIST and assign each point to the one that contains its coordinates. For each of these subcubes that contains a point, we create a node and add it as a child of the root in the tree (so their height is one), with edge weight $\sqrt{d}$MAXDIST, i.e., the side length of the (parent) cube times $\sqrt{d}/2$. Notice that this is equal to the half of the maximum distance between any two coordinates in the parent cube. Also observe that the number of nodes at height one is at most $n$, since each node contains at least one point. We let the height of these edges be zero. We then repeat this operation on the nodes until *every* cube contains at most a single point. This results in a tree where all leaves are at the same height, the height is at most $H = O(\log(d\Delta))$, and there are at most $n$ nodes in each layer. Moreover, the running time of constructing each layer is $O(nd)$ since for each point we can determine in which subcube it belongs by going over its dimensions. The total running time is thus $O(nd \log(d\Delta))$. The distance between two points $p, q$ in the tree, denoted by TREEDIST$(p, q)$, is the length of the shortest path between $p$ and $q$ in the tree, or equivalently twice the length from one of them to their lowest common ancestor.

## 3 Multi-tree Embedding

Tree embedding is a powerful tool for designing approximation algorithms but it cannot be applied directly to the $k$-means problem. In fact there are simple examples that show that the expected distortion between the squared distances of an $\ell_2$ metric and the TREEDIST is $\Omega(n)$. To overcome this limitation, we use *three* tree embeddings with different random shifts and we define the distance between two points (denoted by MULTITREEDIST) to be the minimum TREEDIST among the distances in the three trees. Interestingly, we show that this suffices to get a significantly better upper bound on the distortion. We refer to this simple procedure as MULTITREEINIT(). We note that the running time of MULTITREEINIT() is asymptotically equal to that of a single tree embedding $O(nd \log(d\Delta))$ since it initializes three tree embeddings.

To analyze the expected distortion, define for any set $S \subseteq P$ and point $p \in P$, MULTITREEDIST$(p, S) = \min_{q \in S}$ MULTITREEDIST$(p, q)$. The proof of the following bounds is provided in the full version.

**Lemma 3.1** *For any point $p$, and set $S$, we have* DIST$(p, S)^2 \leq$ MULTITREEDIST$(p, S)^2$ *and* $\mathbf{E}[\text{MULTITREEDIST}(p, S)^2] \leq O(d^2 \cdot \text{DIST}(p, S)^2)$.

## 4 FASTK-MEANS++ Algorithm

Recall that the classic K-MEANS++ algorithm samples the first center uniformly at random and in the remaining iterations $k$-means++ samples a center from the $D^2$-distribution, where the probability of each point is proportional to the squared distance to its nearest current center. The most expensive operation in this procedure is to update the $D^2$-distribution after each sample. In fact, the probability for a point to be selected may change in every round of the algorithm leading to $n$ updates in each of the $k$ iterations.

Our key idea here is to use the special structure of the the multi-tree embedding to update the $D^2$-distribution with respect to those distances efficiently. This is intuitively possible since in the multi-tree metric every node can change its distance from the current set of centers at most $O(\log(d\Delta))$ times. This is true because in order to decrease the distance between a point $x$ and the set of centers in a single tree embedding, the lowest common ancestor between $x$ and the closest center has to get closer to $x$. The number of times that this can happen is bounded by the height of the tree. Therefore, since the multi-tree embedding consists of three trees of height $O(\log(d\Delta))$, we have that the number of times a point can change its multi-tree distance to the set of opened centers is at most $O(\log(d\Delta))$.

**MULTITREEOPEN and MULTITREESAMPLE.** To describe our algorithm we start by defining the procedures to update the distribution, MULTITREEOPEN, and to compute a sample MULTI-TREESAMPLE. To achieve an efficient running time, both procedures act on a common data structure which consists of the following:

- A weight $w_x$ for each point $x \in P$.

- A node-weighted balanced binary tree with a leaf for each of the $n$ points in $P$. We refer to this tree as the sample-tree so as to not confuse it with the trees in the multi-tree embedding.

- For each node in each of the trees in the multi-tree embedding, a bit saying whether this node is marked.

For notational convenience, let $\text{MULTITREEDIST}(x, \emptyset)^2 = M$ for any point $x \in P$, where $M = 16d \cdot \text{MAXDIST}^2$ is chosen to be an upper bound of $\text{MULTITREEDIST}(p, q)^2$ for any two points $p$ and $q$. If we let $S$ be the set of opened points (using calls to $\text{MULTITREEOPEN}$), the data structure will satisfy the following invariants:

1. For every $x \in P$, $w_x = \text{MULTITREEDIST}(x, S)^2$.

2. The weight of each node in the sample-tree equals the sum of the weights of the points corresponding to the leaves in its subtree.

3. A node $v$ in a tree $T$ in the multi-tree embedding is marked if there is a point in its subtree that has been opened, i.e., is in $S$; otherwise it is in unmarked.

So the data structure is initialized (when $S = \emptyset$) by setting all weights $(w_x)_{x \in P}$ to $M$; setting the weight of each node in the sample-tree to $M$ times the number of points in its subtree; and by letting all nodes in the trees of the multi-tree embedding to be unmarked. In addition, for each tree $T$ in the multi-tree embedding and for each node $v$ in $T$, we compute the set $P_T(v) \subseteq P$ of points in its subtree. Note that the initialization of the weights and the sample-tree run in time $O(n)$ whereas the initialization of the unmarked notes and the sets $P_T(v)$ can be computed in time $O(n \log(d\Delta))$ by traversing the trees in the multi-tree embedding of height $O(\log(d\Delta))$. The total runtime of the initialization is thus $O(n \log(d\Delta))$.

We proceed to describe the procedure $\text{MULTITREEOPEN}$ that opens a new point $x$ and updates the data structure to satisfy the invariants. We then describe the simpler procedure $\text{MULTITREESAMPLE}$ which samples a point $x$ with probability $w_x / (\sum_{y \in P} w(y))$, i.e., from the $D^2$-distribution with respect to the multi-tree distances.

The description of $\text{MULTITREEOPEN}$ is given in Algorithm 1. When the tree embedding is not clear from the context, we use the notation $\text{TREEDIST}_T$ to denote the distances given by the tree embedding $T$. We now verify the invariants and give some intuition of the procedure. Let $S$ be the set of opened centers prior to this call to $\text{MULTITREEOPEN}(x)$ and let $T$ be a tree in the multi-tree embedding. When considering $T$, $\text{MULTITREEOPEN}$ starts in the leaf $v_0$ of $T$ that $x$ belongs to. It then traverses the tree towards the root forming a path $v_0, v_1, \dots, v_\ell$ of nodes so that $v_\ell$ is either the root or its parent is already marked. The subtrees of these vertices are exactly those that contain $x$ but no other point in $S$, and so Step 4 guarantees the third invariant. Now a key observation is that $\text{TREEDIST}_T(y, S \cup \{x\}) < \text{TREEDIST}_T(y, S)$ for exactly those points $y$ in $P_T(v_\ell)$. This holds because in order to decrease the distance be-

---

**Algorithm 1** $\text{MULTITREEOPEN}$

---

**Input:** A point $x \in P$

1: **for** each tree $T$ in the multi-tree embedding **do**
2:     Let $v_0$ be the leaf of $T$ that $x$ belongs to.
3:     Traverse the tree towards the root forming a path $v_0, v_1, \dots, v_\ell$ until either $v_\ell$ is the root or the parent of $v_\ell$ is marked.
4:     Mark $v_0, \dots, v_\ell$.
5:     **for** each point $y$ in $P_T(v_\ell)$ **do**
6:         **if** $\text{TREEDIST}_T(y, x)^2 < w_y$ **then**
7:             $w_y \leftarrow \text{TREEDIST}_T^2(y, x)$
8:             Traverse the sample-tree from the leaf corresponding to $y$ to the root to update the node-weights that depend on $w_y$.
9:         **end if**
10:     **end for**
11: **end for**

---

tween a point $y$ and the set of centers, with respect to the tree embedding $T$, the lowest common ancestor in $T$ between y and the closest center must get closer. $\text{MULTITREEOPEN}$ considers each of these points and updates $w_y$ if $\text{TREEDIST}_T(y, x) < w_y$. Since the procedure considers all three trees in the multi-tree embedding this guarantees the first invariant, i.e., that $w_y = \text{MULTITREEDIST}(y, S \cup \{x\})^2$ for every $y \in P$ at the end of the procedure. The second invariant is guaranteed by Step 8 which updates all the nodes in the sample-tree so as to satisfy that invariant. $\text{MULTITREEOPEN}$ therefore updates the data structure to satisfy the invariants. As the distance from a point $x$ to the centers is updated $O(\log(d\Delta))$ times and each time the sample-tree is

updated in time $O(\log n)$ (its height), we have the following running time (see the full version for a formal argument).

**Lemma 4.1** *The running time of opening any set $S$ of $k$ points (using calls to* MULTITREEOPEN*) is $O(n \log(d\Delta) \log n)$.*

Having described how to open a new center, we proceed to describe the simpler algorithm for generating a sample. The pseudo-code of MULTITREESAMPLE is given in Algorithm 2. MULTITREESAMPLE traverses the sample-tree from the root to a random leaf by, at each intermediate node, randomly choosing one of its two children proportional to its weight. As the weight of each node in the sample-tree, equals the sum of weights of the points in its subtree (by the second invariant), this guarantees that a point $x$ is sampled with probability $w_x / \sum_{y \in P} w_y$, i.e., proportional to its weight. By the first invariant, this corresponds to sampling from the $D^2$-distribution with respect to the multi-tree distances[8]. Furthermore, the running time of MULTITREESAMPLE is $O(\log n)$ since the height of the sample-tree is $O(\log n)$. (Recall that the sample-tree is a balanced binary tree with $n$ leafs and is thus of height $O(\log n)$. Recall also that the sample-tree is a different tree from the tree embeddings) We summarize these properties of MULTITREESAMPLE in the following lemma.

---

**Algorithm 2** MULTITREESAMPLE

1: Let $v$ be the root of the sample-tree.
2: **while** $v$ is not a leaf **do**
3:   Let $w(L)$ and $w(R)$ be the weight of its left and right child, respectively.
4:   Update $v$ to be its left child with probability $\frac{w(L)}{w(L)+w(R)}$ and to be its right child with remaining probability $\frac{w(R)}{w(L)+w(R)}$.
5: **end while**
**Output:** the point $x$ corresponding to the leaf $v$.

---

**Algorithm 3** FASTK-MEANS++

**Input:** Set of points $P$, number of centers $k$.
1: Set $S \leftarrow \emptyset$
2: MULTITREEINIT ()
3: **while** $|S| < k$ **do**
4:   $x \leftarrow$ MULTITREESAMPLE()
5:   $S \leftarrow S \cup x$
6:   MULTITREEOPEN$(x)$
7: **end while**
**Output:** $S$

---

**Lemma 4.2** *Let $S$ be the set of opened centers (using calls to* MULTITREEOPEN*). Then* MULTITREESAMPLE *runs in time $O(\log n)$ and each point $x \in P$ is output with probability* $\frac{\text{MULTITREEDIST}(x,S)^2}{\sum_{y \in P} \text{MULTITREEDIST}(y,S)^2}$ .

**FASTK-MEANS++.** We can now present a fast algorithm for the $k$-means problem (see Algorithm 3) that samples each center from the $D^2$-distribution with respect to the distances given by the multi-tree embedding. In the next section we show how to adapt the procedure so as to sample from the original $D^2$-distribution by using rejection sampling. The running time directly follows from that, the time to initialize the multi-tree embedding is $O(nd \log(d\Delta))$, the time to initialize the data structure used by MULTITREEOPEN and MULTITREESAMPLE is $O(n \log(d\Delta))$, the total running time of MULTITREEOPEN is $O(n \log(d\Delta) \log n)$ (Lemma 4.1) and the running time of each call to MULTITREESAMPLE is $O(\log n)$ (Lemma 4.2).

**Corollary 4.3** *The running time of* FASTK-MEANS++ *is $O(nd \log(d\Delta) + n \log(d\Delta) \log n)$.*

## 5 Rejection Sampling Algorithm

In this section we present an algorithm, REJECTIONSAMPLING, that efficiently samples arbitrarily close to the $D^2$-distribution in the original metric. The algorithm is rather simple and its pseudo-code is given in Algorithm 4. The main idea is to use the multi-tree embedding to sample candidate centers but then adjust the sampling probability using rejection sampling.

As for K-MEANS++, the first center that we pick is chosen uniformly at random among all the points. For the rest of the $k - 1$ centers, the idea is to sample a point $x$ using MULTITREESAMPLE, i.e.,

form the $D^2$-distribution with respect to the multi-tree distances. Then we open $x$ as a new center with probability proportional to its actual distance to the set of centers in the original metric over the distance in the multi-tree embedding. We repeat this procedure until we pick the rest of the $k-1$ centers. Interestingly, this rejection-sampling procedure guarantees that we sample each of the centers according to the actual $D^2$-distribution. However, the running time of this procedure is of $\Omega(k^2)$ since, for each point $x$ that we sample from the multi-tree, we have to find the closest open center which takes time $\Omega(k)$. In order to improve this running time, we use an approximate nearest neighbor data structure to approximate the distance between $x$ and the closest open center. This enables us to improve the running time to be near linear. The data structure that we use is based on the locality-sensitive hash (LSH) functions developed for Euclidean metrics [2]. We only need to slightly modify their data structure to guarantee monotonicity as we explain in the full version.

**Theorem 5.1 (LSH data structure)** *For any set $P$ of $n$ points in $R^d$ and any parameter $c > 1$, there exists a data structure with operations Insert and Query that, with probability at least $1 - 1/n$, have the following guarantees: (i) Insert(p):* Inserts *point $p \in P$ to the data structure in time* $O\left(d\log(\Delta) \cdot (n\log(\Delta))^{O(1/c^2)}\right)$. *(ii) Query(p): Returns a point $q$ that has been inserted into the data structure that is at distance at most $c \cdot \delta$ from $p$, where $\delta$ is the minimum distance from $p$ to a point inserted to the data structure. The query time is* $O\left(d\log(\Delta) \cdot (n\log(\Delta))^{O(1/c^2)}\right)$.
*Furthermore, the data structure is* monotone under insertions: *the distance between $p$ and Query(p) is non-increasing after inserting more points.*

We say that the data structure is successful if the above guarantees hold. By the theorem statement, we know that the data structure is successful with probability at least $1 - 1/n$. The small failure probability will not impact the expected cost of our solution[9]. We therefore *assume throughout the analysis that our data structure is successful.* In Algorithm 4 we present the pseudocode for our algorithm.

In the REJECTIONSAMPLING algorithm (Algorithm 4), the probability on Line 5 is not defined for the case that $S$ is an empty set, i.e., the first iteration of the loop. In this case we assume that this probability is one and the sampled element will be added to $S$. We start be presenting a few properties of REJECTIONSAMPLING algorithm. We show that the expected number of the times that the loop (Line 3) repeats is $O(c^2 d^2 k)$. To that end, we first show that the probability of opening a center in $x$ in any iteration is independent of the MULTITREE embedding and only depends on the LSH data structure. This holds, intuitively, because when we sample a point $x$ by calling MULTITREESAMPLE() we then decide to add it based on the distance to

---

**Algorithm 4** REJECTIONSAMPLING

**Input:** Set of points $P$, number of centers $k$
1: Set $S \leftarrow \emptyset$
2: MULTITREEINIT ()
3: **while** $|S| < k$ **do**
4:    $x \leftarrow$ MULTITREESAMPLE()
5:    **With probability** $\min\{1, \frac{\text{DIST}(x,\text{Query}(x))^2}{c^2 \cdot \text{MULTITREEDIST}(x,S)^2}\}$ **do**
6:       $S \leftarrow S \cup x$
7:       MULTITREEOPEN ($x$)
8:       Insert(x)
9: **end while**
**Output:** $S$

---

the point reported by the LSH data structure which removes the dependency on MULTITREEINIT. Specifically, each point $x$ is first sampled w.p. $\frac{\text{MULTITREEDIST}(x,S)^2}{\sum_{y \in P} \text{MULTITREEDIST}(y,S)^2}$ and then added to set $S$ w.p. $\frac{\text{DIST}(x,\text{Query}(x))^2}{c^2 \cdot \text{MULTITREEDIST}(x,S)^2}$. Therefore, the probability of adding $x$ to $S$ is proportional to $\text{DIST}(x, \text{Query}(x))^2$ and we get (see the full version for a formal proof):

**Lemma 5.2** *The probability of inserting a point $x$ to set $S$ in* REJECTIONSAMPLING *algorithm is independent of* MULTITREEINIT *and is equal to $1/n$ for the first iteration and $\frac{\text{DIST}(x,Query(x))^2}{\sum_{y \in P} \text{DIST}(y,Query(y))^2}$ for other iterations.*

| Algorithm | $k = 100$ | $k = 500$ | $k = 1000$ | $k = 2000$ | $k = 3000$ | $k = 5000$ |
|---|---|---|---|---|---|---|
| FASTK-MEANS++ | 1.0x | 1.0x | 1.0x | 1.0x | 1.0x | 1.0x |
| REJECTIONSAMPLING | 1.04x | 1.09x | 1.04x | 1.07x | 1.01x | 1.28x |
| K-MEANS++ | 0.66x | 3.11x | 6.58x | 15.26x | 18.58x | 42.64x |
| AFKMC2 | 0.89x | 1.88x | 3.80x | 8.5x | 16.61x | 38.7x |

Table 1: Running time of the algorithms divided by the running time of FASTK-MEANS++ for the KDD-Cup dataset. This shows the speed-up that we achieve compared to the K-MEANS++ and AFKMC2.

| Algorithm | $k = 100$ | $k = 500$ | $k = 1000$ | $k = 2000$ | $k = 3000$ | $k = 5000$ |
|---|---|---|---|---|---|---|
| FASTK-MEANS++ | 1.0x | 1.0x | 1.0x | 1.0x | 1.0x | 1.0x |
| REJECTIONSAMPLING | 0.99x | 1.03x | 0.98x | 1.03x | 1.04x | 1.04x |
| K-MEANS++ | 0.76x | 4.55x | 8.89x | 16.98x | 23.03x | 46.26x |
| AFKMC2 | 0.62x | 1.02x | 1.35x | 2.81x | 4.98x | 8.71x |

Table 2: Running time of the algorithms divided by the running time of FASTK-MEANS++ for the Song dataset. This shows the speed-up that we achieve compared to the K-MEANS++ and AFKMC2.

The main ingredient in the running time analysis is to bound the number of repetitions of the loop (Line 3). This is roughly done by arguing that the probability that we add an element $x$ to $S$ after its sampled using MULTITREESAMPLE is $\Omega(\frac{1}{c^2 d^2})$ in expectation. Indeed, from Lemma 3.1 we expect that $\text{MULTITREEDIST}(x, S)^2 \leq O(d^2 \text{DIST}(x, \text{Query}(x))^2)$, so $\frac{1}{\Omega(c^2 d^2)} \leq \frac{\text{DIST}(x,\text{Query}(x))^2}{c^2 \cdot \text{MULTITREEDIST}(x,S)^2}$. Therefore the probability of passing Line 5 is at least $\frac{1}{\Omega(c^2 d^2)}$. It follows that, in expectation, $O(c^2 d^2 k)$ repetitions suffices to add $k$ points to $S$. The formal proof is presented in the full version.

**Lemma 5.3** *The expected number of the times that the loop (Line 3) is repeated is of $O(c^2 d^2 k)$.*

Putting the discussed ingredients and the approximation ratio analysis together, we get the following result, the proof is presented in the full version.

**Theorem 5.4** *For any constant $c > 1$, with probability at least $(1 - 1/n)$ REJECTIONSAMPLING always samples points $x$ that are at most a factor $c^2$ away from the $D^2$-distribution, its expected running time is $O\left(n \log(d\Delta)(d + \log n) + kc^2 d^3 \log(\Delta) \cdot (n \log(\Delta))^{O(1/c^2)}\right)$, and it returns a solution that in expectation is a $O(c^6 \log k)$-approximation of the optimal solution.*

We remark that the runtime can be improved in the case of a large $d$ by first applying a dimensionality reduction [7, 21] that reduces the dimension of the input points to $O(\log n)$ in time $O(nd \log n)$ and maintains the cost of any clustering up to a constant factor. These works actually prove that the dimension can be reduced to $O(\log k)$. However, by using $O(\log n)$ our algorithm can output the solution for *all* $k = 1, 2, \ldots, n$ in near-linear running time $\widetilde{O}\left(nd + n \log \Delta + c^2 k \log(\Delta)(n \log \Delta)^{O(1/c^2)}\right)$ (where $\widetilde{O}$ suppresses logarithmic terms in $n$) while maintaining the same asymptotic approximation guarantee as K-MEANS++. Selecting $\varepsilon = O(1/c^2)$ then yields the following

**Corollary 5.5** *For $\varepsilon > 0$, there is an $O_\varepsilon(\log k)$-approximation algorithm for the k-means problem with a running time of $\widetilde{\Theta}(nd + (n \log(\Delta))^{1+\varepsilon})$.*

## 6 Empirical Evaluation

In this section we empirically validate out theoretical results by comparing our algorithms FASTK-MEANS++ and REJECTIONSAMPLING (see the details on how we set the parameters for LSH in the full version ) with the following two baselines:
K-MEANS++ algorithm: Perhaps the most commonly used algorithm in this field. It samples $k$ points according to the $D^2$-distribution.
AFKMC2 algorithm: A recent result [4] based on random walks that improves the running time of the K-MEANS++ algorithm while maintaining a (weaker) theoretical guarantee on the solution quality.

| Algorithm | $k = 100$ | $k = 500$ | $k = 1000$ | $k = 2000$ | $k = 3000$ | $k = 5000$ |
|---|---|---|---|---|---|---|
| FASTK-MEANS++ | 1.0x | 1.0x | 1.0x | 1.0x | 1.0x | 1.0x |
| REJECTIONSAMPLING | 0.98x | 1.26x | 1.17x | 1.07x | 1.0x | 0.95 |
| K-MEANS++ | 0.89x | 4.78x | 8.92x | 14.18x | 23.57x | 36.69 |
| AFKMC2 | 0.76x | 0.77x | 1.12x | 1.15x | 1.54x | 2.56x |

Table 3: Running time of the algorithms divided by the running time of FASTK-MEANS++ for the `Census` dataset. This shows the speed-up that we achieve compared to the K-MEANS++ and AFKMC2.

| Algorithm | $k = 100$ | $k = 500$ | $k = 1000$ | $k = 2000$ | $k = 3000$ | $k = 5000$ |
|---|---|---|---|---|---|---|
| FASTK-MEANS++ | 30335 | 5771 | 2957 | 1582 | 1070 | 640 |
| REJECTIONSAMPLING | 29243 | 5857 | 2999 | 1581 | 1095 | 642 |
| K-MEANS++ | 24552 | 5128 | 2695 | 1423 | 968 | 562 |
| AFKMC2 | 25598 | 5384 | 2883 | 1512 | 1045 | 622 |
| UNIFORMSAMPLING | 148594 | 51692 | 26199 | 15927 | 13922 | 10017 |

Table 4: Costs of the solutions produced by the algorithm for `KDD-Cup` dataset for various values of $k$. All the numbers are scaled down by a factor $10^3$.

**Datasets, Experiments, and Setup**   We ran our algorithms on three classic datasets from UCI library [15]: `KDD-Cup` [12] ($311,029$ points of dimension 74) and `song` [8] ($515,345$ points of dimension 90) `Census` [19] ($2,458,285$ points of dimension 68). We did not apply any dimensionality reduction technique for any of the algorithms; all the considered data sets is of small dimension. We compare the quality of the clustering, i.e., the cost of the objective function, along with their running times. For the AFKMC2 algorithm, we used the code provided by the authors with the same parameter suggested there, i.e., $m = 200$.[10] The algorithms were run on a standard desktop computer.

**Discussion**   Our results show that the algorithms we propose are much faster than both the baselines, i.e., K-MEANS++ and AFKMC2, as $k$ grows. For large $k = 5000$, it is an order of magnitude faster than both K-MEANS++ and AFKMC2. Moreover, the running time of our algorithms is already significantly faster than both baselines for moderate values of $k$ such as $k = 500$ for `KDD-Cup` and $k = 1000$ for `Song` and `Census`. We refer to Tables 1, 2, and 3 for more details.

Importantly, we achieve this improvement in the running time without making any significant sacrifice to solution quality from both a theoretical and experimental perspective. While the solution quality is sometimes worse by 10-15% for small $k$, the $k$-means costs of the solutions produced by FASTK-MEANS++ and REJECTIONSAMPLING algorithms are comparable (overall almost the same) with the baselines for all the experiments for moderate values of $k \geq 1000$. This is in contrast to the simplest seeding algorithm UNIFORMSAMPLING which selects the $k$ centers uniformly at random from the input data set. While UNIFORMSAMPLING clearly provides for a very fast seeding algorithm, it does so by significantly deteriorating the solution quality. This can e.g. be seen in our results for the `KDD-Cup` dataset where UNIFORMSAMPLING consistently gives solutions of much worse quality. For more details, see Tables 4, 5, and 6 where the solution costs are given. The variance along with experimental setting is reported in the full version.

# 7   Conclusions

In this paper we present new efficient algorithms for $k$-means++ seeding. Our algorithms outperform previous work as $k$ grows and come with strong theoretical guarantees. Interesting avenues for future work are to develop efficient distributed algorithms for the same problem and to prove lower bounds on the running time.

| Algorithm | $k = 100$ | $k = 500$ | $k = 1000$ | $k = 2000$ | $k = 3000$ | $k = 5000$ |
|---|---|---|---|---|---|---|
| FASTK-MEANS++ | 21898668 | 16732379 | 14987614 | 13477854 | 12691185 | 11628744 |
| REJECTIONSAMPLING | 21743137 | 16851767 | 15024812 | 13558210 | 12720314 | 11654493 |
| K-MEANS++ | 21583261 | 16409834 | 14746899 | 13395052 | 12480900 | 11496421 |
| AFKMC2 | 21596184 | 16344430 | 14750601 | 13246450 | 12450688 | 11476712 |
| UNIFORMSAMPLING | 23255642 | 17919981 | 16373134 | 14579718 | 13934375 | 12938255 |

Table 5: Costs of the solutions produced by the algorithms for the Song dataset for various values of $k$. All the numbers are scaled down by a factor $10^5$.

| Algorithm | $k = 100$ | $k = 500$ | $k = 1000$ | $k = 2000$ | $k = 3000$ | $k = 5000$ |
|---|---|---|---|---|---|---|
| FASTK-MEANS++ | 17304 | 9820 | 7883 | 6326 | 5625 | 4868 |
| REJECTIONSAMPLING | 17735 | 9970 | 8031 | 6432 | 5644 | 4893 |
| K-MEANS++ | 18498 | 9585 | 7812 | 6254 | 5561 | 4815 |
| AFKMC2 | 17242 | 9844 | 7710 | 6272 | 5595 | 4838 |
| UNIFORMSAMPLING | 19912 | 10630 | 8678 | 6880 | 6120 | 5228 |

Table 6: Costs of the solutions produced by the algorithm for Census dataset for various values of $k$. All the numbers are scaled down by a factor $10^4$.

## Broader Impact

Our work focuses on speeding-up the very popular K-MEANS++ algorithm for clustering. The K-MEANS++ algorithm is used in a variety of domains and is an important tool for extracting information, compressing data, or unsupervised classification tasks. Our result shows that one can obtain a much faster implementation of the $k$-means++ algorithm while preserving its approximation guarantees both in theory and in practice. Therefore, we expect that our new algorithm could have impact in several domains in which clustering plays an important role. A broader concrete impact in society is harder to predict since this is mainly fundamental research.

## Acknowledgments and Disclosure of Funding

The last author is supported by the Swiss National Science Foundation project 200021-184656 "Randomness in Problem Instances and Randomized Algorithms."

## Footnotes

[4]We note that a similar running time can be achieved also via coresets [16, 11] but it is challenging to go below the $\Omega(k^2)$ barrier.

[5]While the large $k$ setting is not the most studied setting, it still has many practical applications. For instance in spam and abuse [23, 25], near-duplicate detection [17], compression or reconciliation tasks [24]. Furthermore the large $k$ case is very interesting from a theoretical perspective and it gained attention in recent years [9].

[6]This can be done in time $O(nd)$, by selecting any point and by computing the maximum distance between that point and any other point in the dataset. Then multiply this distance by 2.

[7]Notice that this does not effect the distance between any two points and therefore the cost of any solution.

[8]We remark that the idea of sampling in this way from a tree has been used in the context of constructing a coreset in [1] (however, their tree depends on a partition of the data and is not necessarily balanced).

[9]To be completely formal: if we repeat our algorithm for $\log_n(4n\Delta^2)$ times, then we know that with probability at least $1 - 1/(4n\Delta^2)$ one of the runs is with a successful data structure. As squared-distances are at most MAXDIST$^2$ and at least MAXDIST$^2/(2\Delta)^2$, the total cost of a solution with a single opened center is at most $n \cdot$ MAXDIST$^2$. Therefore, the small failure probability of $1/(4n\Delta^2)$ will not have a measurable impact on the expected cost of the best found clustering.

[10]$m$ is the number of steps in the random walk.

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
