[Supplementary Material]

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

*Proof.* We start by showing that $\text{DIST}(p,S)^2 \leq \text{MULTITREEDIST}(p,S)^2$. We prove this for every single tree, which implies the result. In particular, we show that for any tree and for any two points $p, q \in P$, we have $\text{DIST}(p,q) \leq \text{TREEDIST}(p,q)$. Assume that the lowest common ancestor of $p, q$ is at height $i$. Therefore in each dimension, they differ at most by $\frac{2 \cdot \text{MAXDIST}}{2^i}$ since the side length of the cube at this height is $\frac{2 \cdot \text{MAXDIST}}{2^i}$, therefore

$$\text{DIST}(p,q) \leq \sqrt{d} \cdot \frac{2 \cdot \text{MAXDIST}}{2^i}.$$

Moreover, $\text{TREEDIST}(p,q)$ is defined as the length of the shortest path between them and the length of the edge at height $j$ is $\sqrt{d} \cdot \frac{\text{MAXDIST}}{2^j}$ for $0 \leq j < H$. So

$$\text{TREEDIST}(p,q) = 2 \sum_{i \leq j < H} \sqrt{d} \cdot \frac{\text{MAXDIST}}{2^j} = 2\sqrt{d} \cdot \text{MAXDIST} \sum_{i \leq j < H} 2^{-j} \geq \sqrt{d} \cdot \frac{2 \cdot \text{MAXDIST}}{2^i}.$$

Therefore, $\text{DIST}(p,q) \leq \sqrt{d} \cdot \frac{2 \cdot \text{MAXDIST}}{2^i} \leq \text{TREEDIST}(p,q)$, which concludes the proof of the first part of the lemma.

Now we focus on the second part of the lemma, i.e., $\mathbf{E}[\text{MULTITREEDIST}(p,q)^2] \leq O(d^2 \cdot \text{DIST}(p,q)^2)$. Let $p = (p_1, \ldots, p_d)$ and $q = (q_1, \ldots, q_d)$ be two points in $\mathbb{R}^d$. We first analyze the probability that these two points are separated at a certain height $i$ in a single tree. In a single tree, two points are separated at height $i$ if they are separated in at least one of the coordinates. The probability that $p$ and $q$ are separated in the $j$-th dimension is at most $\frac{|p_j - q_j|}{\frac{2 \cdot \text{MAXDIST}}{2^i}}$. Therefore, if we let $s_i$ denote the probability that they are separated at height $i$ (but at no smaller height), then by the union bound $s_i \leq \sum_{1 \leq j \leq d} \frac{|p_j - q_j|}{\frac{2 \cdot \text{MAXDIST}}{2^i}} \leq \sqrt{d} \cdot \frac{2^i \text{DIST}(p,q)}{2 \cdot \text{MAXDIST}}$, where the second inequality holds because $\sum_{1 \leq j \leq d} |p_j - q_j| \leq \sqrt{d} \cdot \text{DIST}(p,q)$. So the probability that they are separated at height $i$ or before is at most

$$\sum_{0 \leq j \leq i} s_j \leq \sum_{0 \leq j \leq i} \sqrt{d} \cdot \frac{2^j \text{DIST}(p,q)}{2 \cdot \text{MAXDIST}} \leq \sqrt{d} \cdot \frac{2^{i+1} \text{DIST}(p,q)}{2 \cdot \text{MAXDIST}}.$$

Notice that, as before, we also have that $\text{TREEDIST}(p,q) = 2\sqrt{d} \cdot \text{MAXDIST} \cdot \sum_{i \leq j < H} 2^{-j} \leq \sqrt{d} \cdot \frac{4 \cdot \text{MAXDIST}}{2^i}$ in the case that $p,q$ are separated at height $i$. Now recall that $\text{MULTITREEDIST}$ is the minimum distance among all the three tree embeddings, so it is enough that the two points are separated at height $i$ in a single tree to be at this distance (in the other two trees they can be separated in a level closer to the root). There are three ways to select the tree of minimum distance and so

$$
\begin{aligned}
\mathbf{E}\big[\text{MULTITREEDIST}(p,q)^2\big] &\leq 3 \sum_{0 \leq i < H} s_i \left( \sum_{0 \leq j \leq i} s_j \right)^2 \cdot \left( \sqrt{d} \cdot \frac{4 \cdot \text{MAXDIST}}{2^i} \right)^2 \\
&\leq 3 \sum_{0 \leq i < H} s_i \left( \sqrt{d} \cdot \frac{2^{i+1} \text{DIST}(p,q)}{2 \cdot \text{MAXDIST}} \right)^2 \cdot \left( \sqrt{d} \cdot \frac{4 \cdot \text{MAXDIST}}{2^i} \right)^2 \\
&= 3 \sum_{0 \leq i < H} s_i 16 d^2 \text{DIST}(p,q)^2 = 48 d^2 \text{DIST}(p,q)^2 \sum_{0 \leq i < H} s_i \\
&\leq 48 d^2 \text{DIST}(p,q)^2 = O(d^2 \text{DIST}(p,q)^2),
\end{aligned}
$$

where the last inequality holds because $\sum s_i \leq 1$ since the $s_i$'s denote the probabilities of mutually disjoint events. $\qquad \square$

## B  Proof Omitted from Section 4

**Lemma 4.1** *The running time of opening any set $S$ of $k$ points (using calls to $\text{MULTITREEOPEN}$) is $O(n \log(d\Delta) \log n)$.*

*Proof.* As the multi-tree embedding consists of three trees, it is sufficient to analyze the running time of the for-loop at Step 1 for a single tree $T$. In a single call to $\text{MULTITREEOPEN}$ we have that Steps 2- 4 runs in time $O(\log(d\Delta))$ since each tree in the multi-tree embedding has depth at most $H = \log(d\Delta)$. Hence the total running time for these steps across the $k$ calls to $\text{MULTITREEOPEN}$ is $O(k \log(d\Delta))$.

The for-loop at Step 5 can be implemented as follows. Observe that the distance $\text{TREEDIST}_T(x,y)$ for a point $y \in P_T(v_0)$ equals 0. Furthermore, for $i = 1, 2, \ldots, \ell$ and a point $y \in P_T(v_i) \setminus P_T(v_{i-1})$, the distance $\text{DIST}_T(x,y)$ equals twice the length of the path in $T$ from $v_0$ to $v_i$. We can thus calculate all relevant distances in time $O(\log(d\Delta))$ (and in time $O(k \log(d\Delta))$ across all $k$ calls).

The weights of the points can then be updated by first considering the points in $P_T(v_0)$, then those in $P_T(v_1) \setminus P_T(v_0)$, and so on until the points in $P_T(v_\ell) \setminus P_T(v_{\ell-1})$. The running time of Step 7 is thus $O(\sum_{i=0}^{\ell} |P_T(v_i)|)$. Moreover, as each execution of Step 8 takes time $O(\log n)$ (since the sample-tree is balanced binary tree with $n$ leaves and thus of height $O(\log n)$), the running time of this step is $O(\sum_{i=0}^{\ell} |P_T(v_i)| \cdot \log n)$. Now a key point is that a node in a tree in the multi-tree embedding can only be marked once. Therefore, using that $\sum_v |P_T(v)| = O(n \log(d\Delta))$, we have that the total running time of the for-loop is $O(n \log(d\Delta) \log n)$. The total running time of the $k$ calls to $\text{MULTITREEOPEN}$ is therefore $O(k \log n + n \log(d\Delta) \log n)$. $\qquad \square$

## C  Proofs Omitted from Section 5

**Lemma 5.2** *The probability of inserting a point $x$ to set $S$ in $\text{REJECTIONSAMPLING}$ algorithm is independent of $\text{MULTITREEINIT}$ and is equal to $1/n$ for the first iteration and $\frac{\text{DIST}(x,Query(x))^2}{\sum_{y \in P} \text{DIST}(y,Query(y))^2}$ for other iterations.*

*Proof.* The proof is by induction on the size of the set $S$. The base case is clear since, as aforementioned, in Line 5 we accept the point with probability one and $\text{MULTITREESAMPLE}$ () returns each point with probability $1/n$. Now assume that lemma holds until $\ell$ centers are added (i.e., $|S| = \ell$). It follows from the induction hypothesis the current set $S$ is independent from the MULTITREE initialization since all the elements added so far are independent. By the assumption that

the data structure is successful, the above minimum on Line 5 is always attained by the second term, i.e., we have $\frac{\text{DIST}(x, \text{Query}(x))^2}{c^2 \cdot \text{MULTITREEDIST}(x, S)^2} \leq 1$. Indeed, $\text{DIST}(x, \text{Query(x)})^2 \leq c^2 \cdot \text{DIST}(x, S_i)^2 \leq c^2 \cdot \text{MULTITREEDIST}(x, S_i)^2$, where the first inequality is by the success of the data structure and the second inequality is by the fact that the multi-tree embedding only increases distances (see Lemma 3.1).

Therefore in REJECTIONSAMPLING algorithm, the probability that a point $x$ is sampled is

$$\frac{\text{MULTITREEDIST}(x, S)^2}{\sum_{y \in P} \text{MULTITREEDIST}(y, S)^2} \cdot \frac{\text{DIST}(x, \text{Query}(x)^2)}{c^2 \cdot \text{MULTITREEDIST}(x, S)^2}$$

$$= \frac{\text{DIST}(x, \text{Query}(x))^2}{c^2 \cdot \sum_{y \in P} \text{MULTITREEDIST}(y, S)^2} .$$

Notice that the denominator does not depend on the point $x$ and one can think of it as a constant term. We repeat the sampling process until we pick a point. Therefore, the probability of choosing any point $x$ is $\frac{\text{DIST}(x, \text{Query}(x))^2}{\sum_{y \in P} \text{DIST}(y, \text{Query}(y))^2}$, independent of the the MULTITREE initialization. This completes the inductive step and concludes the proof of the lemma. □

**Lemma 5.3** *The expected number of the times that the loop (Line 3) is repeated is of $O(c^2 d^2 k)$.*

*Proof.* Let $R$ be the random variable that equals the number of times that the loop is repeated. We further divide $R$ into $R_0, \ldots, R_{k-1}$ where, for $i \in \{0, \ldots, k-1\}$, $R_i$ denotes the number of times the loop is executed when the set $S$ of opened centers has size $i$, i.e., when exactly $i$ centers have been opened. Then $R = R_0 + R_1 + \ldots + R_{k-1}$ and, by linearity of expectation,

$$\mathbf{E}[R] = \mathbf{E}[R_0] + \mathbf{E}[R_1] + \ldots + \mathbf{E}[R_{k-1}] .$$

We have $\mathbf{E}[R_0] = 1$ since the first center is selected uniformly at random and it is always opened, i.e., added to $S$. We complete the proof by proving

$$\mathbf{E}[R_i] \leq O(c^2 \cdot d^2) \qquad \text{for } i \in \{1, \ldots, k-1\}.$$

Consider $R_i$ and let $S_i$ be the set containing the first $i$ centers that were opened by the algorithm. We actually prove the stronger statement that $\mathbf{E}[R_i] \leq O(c^2 \cdot d^2)$ no matter the set $S_i$.

Consider an iteration of the loop. First, as argued in the proof of Lemma 5.2, the probability that an iteration of the loop results in adding a point $x$ to the set of opened centers equals $\frac{1}{c^2} \cdot \frac{\sum_{x \in P} \text{DIST}(x, \text{Query}(x))^2}{\sum_{y \in P} \text{MULTITREEDIST}(y, S_i)^2}$. If we let $q$ denote this probability then $\mathbf{E}[R_i] = \sum_{t=1}^{\infty} tq \cdot (1-q)^{t-1}$, which equals $1/q$.

We thus have

$$\mathbf{E}[R_i] = 1/q = c^2 \cdot \frac{\sum_{y \in P} \text{MULTITREEDIST}(y, S_i)^2}{\sum_{x \in P} \text{DIST}(x, \text{Query}(x))^2} ,$$

for a fixed multi-tree embedding.

The lemma now follows from that $\text{DIST}(x, \text{Query}(x))^2 \geq \text{DIST}(x, S_i)^2$ and from Lemma 5.2 which says that the distribution of the random multi-tree embedding is independent from $S_i$. We thus have, by also taking the expectation over the random multi-tree embedding (see Lemma 3.1), that

$$\mathbf{E}[R_i] \leq c^2 \cdot \frac{O(d^2) \sum_{y \in P} \text{DIST}(y, S_i)^2}{\sum_{x \in P} \text{DIST}(x, \text{Query}(x))^2} = O(c^2 \cdot d^2) .$$

□

# D  LSH data structure

In this section we describe the data structure guaranteed by Theorem 5.1. It follows the construction first introduced in [23]. Their construction is based on locality-sensitive hash families:

**Definition D.1 (Locality-sensitive hashing)** *Let $\mathcal{H}$ be a family of hash functions mapping $\mathbb{R}^d$ to some universe $U$. We say that $\mathcal{H}$ is $(R, cR, p_1, p_2)$-sensitive if for any $p, q \in \mathbb{R}^d$ it satisfies the following properties:*

- *If $\|p - q\|_2 \leq R$ then $\Pr_{\mathcal{H}}[h(p) = h(q)] \geq p_1$.*

- *If $\|p - q\|_2 \geq cR$ then $\Pr_{\mathcal{H}}[h(p) = h(q)] \leq p_2$.*

The specific family of hash functions that we use is by [3]. We summarize the main properties of their family in the following theorem. Here, and in the following, we denote by $n$ the size of the data set $P \subseteq \mathbb{R}^d$.

**Theorem D.2 ([3])** *For any $R > 0$ and $c > 1$, there exists a family $\mathcal{H}$ of hash functions for $\mathbb{R}^d$ with the following properties:*

- *$\mathcal{H}$ is $(R, cR, p_1, p_2)$-sensitive with $\frac{\log(1/p_1)}{\log(1/p_2)} = 1/c^2 + o(1)$ and $\frac{1}{\log(1/p_2)} = O(1)$.*

- *The time to compute $h(p)$ for $h \in \mathcal{H}$ and $p \in \mathbb{R}^d$ is $O(dn^{o(1)})$.*

We first describe a data structure for the "gap version". Then we show, using standard arguments, that this gives the data structure as stated in Theorem 5.1.

## D.1  Monotone data structure for gap version

In this section we are going to develop a data structure that is parameterized by $c \geq 1$ (the accuracy) and $R > 0$ (the scale). We refer to it as the $(c, R)$-gap data structure. It is different from the data structure guaranteed by Theorem 5.1 as it only have guarantees that depend on the scaling parameter $R$ (see the statement of Theorem D.3 below).

**Selection of parameters.**   We let $\mathcal{H}$ be the $(R, cR, p_1, p_2)$-sensitive hash family for $\mathbb{R}^d$ given by Theorem D.2. We also let $\delta > 0$ be a parameter of our data structure that determines the probability of failure (and impacts the running time). Other parameters that we use are now determined as follows:

- $\eta = \left(\frac{\delta}{n}\right)^{\frac{3}{1-\rho}}$ where $\rho = \frac{\log(1/p_1)}{\log(1/p_2)}$,

- $m = \frac{\log(1/\eta)}{\log(1/p_2)} = O(\log(1/\eta))$, and

- $\ell = 100 \cdot \log(1/\eta) \cdot (1/\eta)^{\rho}$.

**Description of data structure.**   The data structure is based on $\ell$ hash tables $T_1, T_2, \ldots, T_\ell$ (with linked lists at each entry to deal with collisions). The $\ell$ hash functions $f_1, f_2, \ldots, f_\ell$ for these tables are constructed from $\mathcal{H}$ as follows: for $i \in \{1, 2, \ldots, \ell\}$, $f_i$ is obtained by selecting $m$ *independent samples* $h_{i,1}, h_{i,2}, \ldots, h_{i,m}$ from $\mathcal{H}$. That is, $f_i$ is a $m$-dimensional hash function defined by

$$f_i(p) = [h_{i,1}(p), h_{i,2}(p), \ldots, h_{i,m}(p)] \qquad \text{for } p \in \mathbb{R}^d.$$

We are now ready to define the operations Insert and Query:

- Insert($p$): A point $p \in P \subseteq \mathbb{R}^d$ is inserted in each of the $\ell$ hash tables by appending the point at *the end* of the linked list associated to the entry $T_i[f_i(p)]$ for $i = 1, 2, \ldots, \ell$.
- Query($p$): For each $i \in \{1, 2, \ldots, \ell\}$, let $q_i$ be the first element (if any) in the linked list $T_i[f_i(p)]$ that satisfies $\mathrm{DIST}(p, q_i) \leq cR$. This gives up to $\ell$ candidate points, one for each hash table. Among these candidate points, *output the one with the minimum distance to $p$* (or output none if no candidate point is found in any of the hash tables).

This completes the description of the $c$-NN data structure and we proceed to its analysis.

**Analysis.** We show that the described data structure satisfies the following guarantees:

**Theorem D.3** *For a data set $P \subseteq \mathbb{R}^d$ of $n$ points, the data structure with error parameter $\delta > 0$ satisfies the following guarantees:*

1. *The Insert operation runs in time $O\left(d \cdot (n/\delta)^{O(1/c^2)}\right)$.*

2. *With probability at least $1 - \delta$, the Query operation satisfies the following. Given $p \in P$, if there exists an inserted point within distance $R$ from $p$, then Query(p) returns a point $q$ with $\text{DIST}(p, q) \leq cR$. Moreover, the running time is time is $(n/\delta)^{O(1/c^2)}$.*

*Furthermore, the data structure is* monotone under insertions*: the distance between $p$ and Query(p) is non-increasing after inserting more points.*

Throughout the analysis we assume that $c$ is a large enough constant and that $n$ is sufficiently large. This is motivated by the fact that otherwise a trivial data structure can achieve the bounds claimed by the theorem.

The analyses of the monotonicity property and the running time of the insertion operation are rather immediate:

- The monotonicity property is by definition of the operations Insert and Query. To see that, suppose we run Query(p) for a point $p$. We will argue that inserting any new point $p'$ may not increase the distance $\text{DIST}(p, \text{Query}(p))$. Indeed, when $p'$ is inserted it is appended to the end of the linked-lists $T_i[f_i(p)]$ for $i = 1, 2, \ldots, \ell$. Now when we execute Query(p) the only way that $p'$ will be one of the candidate points $q_1, q_2, \ldots, q_\ell$ is if it, for some $i \in \{1, 2, \ldots, \ell\}$, is the first point in $T_i[f_i(p)]$ within distance $cR$ from $p$. It follows (since insertions are appended at the end of the linked-lists whereas queries inspects the lists from the beginning) that all the the points that were candidates before the insertion of $p'$ are still candidates. Therefore the distance from $p$ to the minimum distance point (of the candidates) can only decrease after inserting a new point $p'$.

- We proceed to analyze the running time of the Insert operation. On the insertion of a point $p \in P$, it is appended to each of the $\ell$ linked lists $T_1[f_1(p)], T_2[f_2(p)], \ldots, T_\ell[f_\ell(p)]$. Appending an element to a linked list takes $O(1)$ time whereas the cost of calculating a single hash $f_i(p)$ is $m$ times the cost of calculating $h(p)$ for a single $h \in \mathcal{H}$, which in turn by Theorem D.2 is $O(dn^{o(1)})$. The running time of an insertion is therefore dominated by the time it takes to calculate the $\ell$ hashes $f_1(p), f_2(p), \ldots, f_\ell(p)$, which by the above arguments takes time

$$\ell \cdot m \cdot O(dn^{o(1)}) = O\left(\log(1/\eta) \cdot (1/\eta)^\rho\right) \cdot O\left(\log(1/\eta)\right) \cdot O(dn^{o(1)})$$
$$= O\left(\log(n/\delta)^2 \cdot (n/\delta)^{3\rho/(1-\rho)}\right) \cdot O(dn^{o(1)})$$
$$= O\left(d \cdot (n/\delta)^{O(1/c^2)}\right),$$

where we used that $c$ is a large enough constant for the last equality.

We proceed to analyze the Query operation which requires a little more work. In order to guarantee that Query returns a nearby point if one exists, we need bound the probability of having a false negative. On the other hand, to bound the running time of the Query operation we need to bound the false positives. The following two lemmas bounds these quantities, starting with the probability of false positives.

**Lemma D.4** *For any $i \in \{1, 2, \ldots, \ell\}$ and two points $p, q \in \mathbb{R}^d$ with $\text{DIST}(p, q) \geq cR$, we have*
$$\Pr[f_i(p) = f_i(q)] \leq \eta.$$

*Proof.* By the independence of $h_{i,1}, h_{i,2}, \ldots, h_{i,m}$, we have
$$\Pr[f_i(p) = f_i(q)] = \Pr_{h \sim \mathcal{H}}[h(p) = h(q)]^m \leq p_2^m,$$

which by the selection of $m$ equals $\eta$. $\qquad\square$

**Lemma D.5** *For any two points $p, q \in \mathbb{R}^d$ with $\mathrm{DIST}(p, q) \leq R$, we have*
$$\Pr[\exists i \mid f_i(p) = f_i(q)] \geq 1 - \eta\,.$$

*Proof.* Similar to the calculations in the proof of the previous lemma, we have
$$\begin{aligned}
\Pr[\exists i \mid f_i(p) = f_i(q)] &= 1 - \Pr[\forall i, f_i(p) \neq f_i(q)] \\
&= 1 - \Pr[f_i(p) \neq f_i(q)]^\ell \\
&\geq 1 - (1 - p_1^m)^\ell\,.
\end{aligned}$$
By the definition of $\rho$, $p_1 = p_2^\rho$ and so
$$\begin{aligned}
\Pr[\exists i \mid f_i(p) = f_i(q)] &\geq 1 - (1 - p_2^{\rho m})^\ell \\
&= 1 - (1 - \eta^\rho)^\ell \\
&\geq 1 - \eta\,,
\end{aligned}$$
where the last inequality is by the selection of $\ell$. $\qquad\square$

Equipped with these two lemmas we are now ready to analyze the Query operaton. Specifically, we have the following corollary:

**Corollary D.6** *Consider a set $P$ of $n$ points in $\mathbb{R}^d$. Then with probability at least $1 - \delta$ we have that the hash functions $f_1, f_2, \ldots, f_\ell$ satisfy the following:*

- *For any $p, q \in P$ with $\mathrm{DIST}(p, q) \geq cR$, we have $f_i(p) \neq f_i(q)$ for all $i \in \{1, 2, \ldots, \ell\}$.*

- *For any $p, q \in P$ with $\mathrm{DIST}(p, q) \leq R$, we have that there is an $i \in \{1, 2, \ldots, \ell\}$ such that $f_i(p) = f_i(q)$.*

Before giving the proof of the corollary, note that the first property implies that we have no false positives. Therefore, the running time of Query is the same as for Insertion: it is dominated by the time to calculate the $\ell$ hash functions which is $O\left(d \cdot (n/\delta)^{O(1/c^2)}\right)$. Moreover, the second property guarantees that we always have a hash collision when there is a nearby point of the query-point $p$. This implies that Query($p$) returns a point $q$ with $\mathrm{DIST}(p, q) \leq cR$ if there is a point within distance $R$ from $p$ that has been inserted. To complete the proof of Theorem D.3 it thus remains to prove the corollary:

*Proof of Corollary.* We show that each of the two properties hold fail probability at most $\delta/2$. The statement then follows by the union bound.

For the first property, there are $\ell$ hash functions and at most $\binom{n}{2} \leq n^2$ pairs $p, q \in P$ such that $\mathrm{DIST}(p, q) \geq cR$. Therefore, by Lemma D.4 and the union bound, we have that the first property fails with probability at most
$$\begin{aligned}
\ell \cdot n^2 \cdot \eta &= (100 \cdot \log(1/\eta) \cdot (1/\eta)^\rho) \cdot n^2 \cdot \eta \\
&= (100 \cdot \log(1/\eta)) \cdot n^2 \cdot \eta^{1-\rho} \\
&= \left(\frac{300}{1 - \rho} \cdot \log(n/\delta)\right) \cdot n^2 \cdot (\delta/n)^3 \\
&\leq \delta/2\,,
\end{aligned}$$
where for the last inequality we used that $n$ and $c$ are large.

For the second property, there are at most $\binom{n}{2} \leq n^2$ pairs $p, q \in P$ such that $\mathrm{DIST}(p, q) \leq R$. So by the union bound and Lemma D.5, we have that the second property fails with probability at most $n^2 \cdot \eta$ which by the above calculations is at most $\delta/2$. $\square$

## D.2 Putting everything together: Proof of Theorem 5.1

The proof of Theorem 5.1 now follows from Theorem D.3 by standard arguments. Again we assume that $c$ is a large constant (since otherwise a trivial data structure will satisfy the properties of the theorem).

Recall that all distances are between $\text{MAXDIST}/(2\Delta)$ and $\text{MAXDIST}$. We make $\log(2\Delta)$ many copies of the gap data structure guaranteed by Theorem D.3. Each of the copies will have an error parameter $\delta = \frac{1}{n\log(2\Delta)}$ and the $i$:th copy will have parameters $(c_i, R_i)$ with $c_i = c/2$ and $R_i = 2^{i-1}\text{MAXDIST}/(2\Delta)$. The operations now work as follows:

- Insert($p$): the point $p \in P$ is inserted into each of the $\log(2\Delta)$ copies of the gap data structure;

- Query($p$): we query the point $p$ in each of the $\log(2\Delta)$ copies and out of the returned points, we return the closest to $p$.

Since the gap data structure of Theorem D.3 is monotone we have that the resulting data structure satisfies monotonicity. That it succeeds with probability at least $1 - 1/n$ follows from the selection of $\delta$ and the union bound over $\log(2\Delta)$ many copies of the gap data structure. Furthermore the guarantees of the query operation (to find a nearby point) is satisfied: let $q$ be the closest point to $p$ and suppose that $2^{i-2} \cdot \text{MAXDIST}/(2\Delta) \le \text{DIST}(p, q) \le 2^{i-1} \cdot \text{MAXDIST}/(2\Delta)$. Then, on Query($p$), the $i$:th copy of the gap data structure is guaranteed to return a point within distance $c/2 \cdot 2^{i-1} \cdot \text{MAXDIST}/(2\Delta) = c \cdot 2^{i-2} \cdot \text{MAXDIST}/(2\Delta) \le c \cdot \text{DIST}(p, q)$ of $p$. Finally the running time of the operations is $\log(2\Delta)$ times the running time of each operation in the gap data structure. Hence, since $\delta = 1/(n\log(2\Delta))$, the running time of the operations is $O\left(\log(2\Delta) \cdot d \cdot \left(n^2 \log(2\Delta)\right)^{O(1/c^2)}\right) = O\left(d\log(\Delta) \cdot (n\log(\Delta))^{O(1/c^2)}\right)$ as required.

## D.3 LSH Parameters in our Experiments

We use the locality sensitive hash families based on $p$-stable distribution introduced by Datar et al. [17]. We set the parameters as follows. We work with one scale, and we set the number of hash functions to be 15. Moreover, we set the collision parameter (referred to as $r$ in [17] to be 10).

# E REJECTIONSAMPLING Algorithm Analysis

In this section we analysis the REJECTIONSAMPLING algorithm. We start by proving approximation guarantee and then stating the main theorem.

## E.1 Analysis of Approximation Guarantee

In this section we prove that REJECTIONSAMPLING has an approximation guarantee of $O(c^6 \log(k))$. Hence, for a fixed $c$, it has the same asymptotic approximation guarantee as the standard implementation of K-MEANS++ but with the advantage that it runs in near-linear time. For simplicity we assume that the LSH data structure is successful throughout the whole analysis. That is, for any $p \in P$, Query($p$) returns a point within distance $c \cdot \delta$ where $\delta$ is the minimum distance from $p$ to a point inserted in the data structure.

Theorem 5.4 says that the probability to sample a center in REJECTIONSAMPLING is very close to the same $D^2$-distribution as in K-MEANS++. At first, it therefore appears rather intuitive that they should have the same approximation guarantee. However, the analysis of K-MEANS++ is rather sensitive to even small perturbations to the probability of sampling a center. Indeed, in a recent paper [11], it was proved that the version of K-MEANS++ where centers are sampled using an approximation of the $D^2$-distribution achieves an approximation guarantee of $O(\log(k)^2)$. To get a tight guarantee of $O(\log(k))$ was raised as an open problem. Our analysis does not resolve this question. Instead we use the additional *monotonicity* property of our LSH data structure (see Theorem 5.4) to circumvent the most technical difficulty of [11]. This allows us to establish the tight asymptotic approximation guarantee of our procedure. Similarly to the proof in [11], our analysis closely follows Dasgupta's

analysis of K-MEANS++ [16]. The main difference is a slight change of the "potential" function (see (1)). However, for the sake of completeness, we reproduce the complete analysis here.

**Notation:** Throughout the proof, we use the following notation. For a set $P' \subseteq P$ of the points and an (ordered) set of centers $S = \{s_1, \ldots, s_i\}$ let

- $\Phi(P', S)$ be the $k$-means cost of data points $P'$ with respect to the centers $S$, i.e.,

$$\Phi(P', S) = \sum_{x \in P'} \text{DIST}(x, S)^2 \, ,$$

- $\Phi_{\text{LSH}}(P', S)$ be the $k$-means cost of data set $P'$ with respect to the centers $S$ when using the assignment given by the LSH data structure, i.e.,

$$\Phi_{\text{LSH}}(P', S) = \sum_{x \in P'} \text{DIST}(x, \text{Query}(x))^2 \, ,$$

where the points of $S$ have been inserted into the data structure in the order $s_1, s_2, \ldots, s_i$.

Furthermore, we denote by $\text{OPT}_i(P)$ the cost of an optimal clustering of the data points $P$ using $i$ centers and we let $C_1^*, C_2^*, \ldots, C_k^*$ be the partition of $P$ into $k$ cluster in a fixed optimal solution (with $k$ centers).

### E.1.1 Two preliminary lemmas

We start our analysis with two preliminary lemmas which are very similar to lemmas in [11], which in turn are based on similar lemmas in the original K-MEANS++ paper [4].

As the first center is chosen uniformly at random in both REJECTIONSAMPLING and K-MEANS++, we can reuse the following statement from the original analysis.

**Lemma E.1 (Lemma** 3.1 **in [4])** *Let $s_1$ denote the first center chosen by* REJECTIONSAMPLING. *For each optimal cluster $C_i^*$,*

$$\mathbf{E}\left[\Phi(C_i^*, \{s_1\}) \mid s_1 \in C_i^*\right] \leq 2 \cdot \text{OPT}_1(C_i^*) \, .$$

For the next lemma, we use that Theorem 5.4 says that a center $s$ is sampled with a probability in $[q/c^2, q \cdot c^2]$ where $q$ denotes the probability that $s$ would be sampled by the $D^2$-distribution. This allows us to use Lemma 5 in the noisy K-MEANS++ analysis:

**Lemma E.2 (Lemma** 5 **in [11])** *Consider* REJECTIONSAMPLING *after at least one center has been opened and let $S \neq \emptyset$ denote the current set of centers. We denote by $s$ the next sampled center. Then for any $S \neq 0$ and any optimal cluster $C_i^*$,*

$$\mathbf{E}\left[\Phi(C_i^*, S \cup \{s\}) \mid S, s \in C_i^*\right] \leq 8c^4 \cdot \text{OPT}_1(C_i^*) \, .$$

### E.1.2 Dasgupta's potential argument

Consider a run of REJECTIONSAMPLING and let $S_i = \{s_1, s_2, \ldots, s_i\}$ denote the first $i$ centers chosen by REJECTIONSAMPLING (for notational convenience, we let $S_0 = \emptyset$). We say that a cluster $C_j^*$ of the optimal solution is covered by $S_i$ if one of its centers is in $C_j^*$. Otherwise we say that this cluster is uncovered. For $i \in \{0, 1, \ldots, k\}$, let $H_i$ and $U_i$ denote the set of all points from $P$ that, with respect to $S_i$, belong to covered and uncovered optimal clusters, respectively. Also let $u_i$ denote the number of uncovered clusters after $i$ centers were opened. Finally, we say that a center $s_i$ is wasted if $s_i \in H_{i-1}$, i.e., if the $i$:th center $s_i$ does not cover a previously uncovered cluster.

The following is an immediate corollary of the two preliminary lemmas; it is Corollary 6 in [11].

**Corollary E.3** *For any $i \in [k]$,*

$$\mathbf{E}\left[\Phi(H_i, S_i)\right] \leq 8c^4 \cdot \text{OPT}_k(P) \, .$$

The above corollary, says that the cost of covered clusters is at most a constant times the cost of an optimal solution. To bound the expected cost of uncovered clusters we use the argument of [16]. It is based on a potential function argument. Define $W_i$ to be the number of wasted centers among the first $i$ centers. Hence $W_i$ equals $i$ minus the number of covered clusters. Further, let

$$\Psi_i = W_i \cdot \frac{\Phi_{\mathrm{LSH}}(U_i, S_i)}{u_i} \,. \tag{1}$$

Our potential $\Psi_i$ is different from the one used in [16] in that we use $\Phi_{\mathrm{LSH}}$ instead of $\Phi$. This is the main difference and it is crucial for our analysis.

For intuition, note that, for $i = 0$, we have no wasted centers and all clusters are uncovered. So $W_0 = 0$ and $u_0 = k$ and $\Psi_0 = 0$. At the other end (for $i = k$), we have that the number of wasted centers equals the number of uncovered clusters, i.e., $W_k = u_k$, and so $\Psi_k$ equals the total cost of uncovered clusters. The definition of $\Psi_i$ allows us to bound this cost step-by-step. In particular, we will bound the expected increase of (1) from $i$ to $i + 1$, i.e., $\mathbf{E}\left[\Psi_{i+1} - \Psi_i\right]$. We emphasize that the analysis is close to a verbatim transcript of that in [16]; it is included for completeness.

In the following, we let $\mathcal{F}_i$ denote the realization of REJECTIONSAMPLING of the first $i$ centers. Any realization $\mathcal{F}_i$ determines e.g. the values of $\Phi_{\mathrm{LSH}}(U_i, S_i)$ and $u_i$.

We consider two cases: when the new center is in an uncovered cluster (Lemma E.4) and when it is in a covered cluster (Lemma E.5).

**Lemma E.4 (Lemma 8 in [16])** *Suppose that the $(i + 1)$:th center $s = s_{i+1}$ is chosen in $U_i$. Then for any $\mathcal{F}_i$*

$$\mathbf{E}\left[\Psi_{i+1} - \Psi_i \mid \mathcal{F}_i, \{s \in U_i\}\right] \leq 0 \,.$$

*Proof.* When $s$ belongs to an uncovered cluster $A$, we have $H_{i+1} = H_i \cup A$, $W_{i+1} = W_i$, $U_{i+1} = U_i \setminus A$, and $u_{i+1} = u_i - 1$. Hence, using the monotonicity of the LSH data structure,

$$
\begin{aligned}
\Psi_{i+1} &= \frac{W_{i+1} \cdot \Phi_{\mathrm{LSH}}(U_{i+1}, S_{i+1})}{u_{i+1}} \\
&\leq \frac{W_i \cdot (\Phi_{\mathrm{LSH}}(U_i, S_i) - \Phi_{\mathrm{LSH}}(A, S_i))}{u_i - 1} \,.
\end{aligned}
$$

Let us bound the cost $\Phi_{\mathrm{LSH}}(A, S_i)$ for a randomly chosen uncovered cluster $A$. Here we use the notation $A(s)$ to denote the uncovered cluster so that $s \in A$. Since a point $s \in U_i$ is sampled proportional to $\Phi_{\mathrm{LSH}}(s, S_i)$

$$
\begin{aligned}
\mathbf{E}&\left[\Phi_{\mathrm{LSH}}(A(s), S_i) \mid \mathcal{F}_i, \{s \in U_i\}\right] \\
&= \sum_A \frac{\Phi_{\mathrm{LSH}}(A, S_i)}{\Phi_{\mathrm{LSH}}(U_i, S_i)} \cdot \Phi_{\mathrm{LSH}}(A, S_i) \\
&\geq \frac{\Phi_{\mathrm{LSH}}(U_i, S_i)}{u_i} \,,
\end{aligned}
$$

where the sum is over the $u_i$ uncovered clusters $A$ and the last inequality is by the Cauchy-Schwarz inequality. Thus, $\mathbf{E}\left[\Psi_{i+1} \mid \mathcal{F}_i, \{s \in U_i\}\right]$ is at most

$$
\begin{aligned}
\frac{W_i}{u_i - 1} &\left(\Phi_{\mathrm{LSH}}(U_i, S_i) - \mathbf{E}\left[\Phi_{\mathrm{LSH}}(A(s), S_i) \mid \mathcal{F}_i, \{s \in U_i\}\right]\right) \\
&\leq \frac{W_i}{u_i - 1} \left(\Phi_{\mathrm{LSH}}(U_i, S_i) - \frac{\Phi_{\mathrm{LSH}}(U_i, S_i)}{u_i}\right) = \Psi_i \,.
\end{aligned}
$$

$\square$

**Lemma E.5 (Lemma 9 in [16])** *Suppose that $(i + 1)$ center $s = s_{i+1}$ is chosen in $H_i$. Then for any $\mathcal{F}_i$, $\Psi_{i+1} - \Psi_i \leq \Phi_{LSH}(U_i, S_i)/u_i$.*

*Proof.* When $s$ is chosen from a covered cluster, we have $H_{i+1} = H_i, U_{i+1} = U_i, u_{i+1} = u_i$, and $W_{i+1} = W_i + 1$. Thus by the monotonicity of our data structure

$$\begin{aligned}
\Psi_{i+1} - \Psi_i &= \frac{W_{i+1} \cdot \Phi_{\text{LSH}}(U_{i+1}, S_{i+1})}{u_{i+1}} - \frac{W_i \cdot \Phi_{\text{LSH}}(U_i, S_i)}{u_i} \\
&\leq \frac{(W_i + 1) \cdot \Phi_{\text{LSH}}(U_i, S_i)}{u_i} - \frac{W_i \cdot \Phi_{\text{LSH}}(U_i, S_i)}{u_i} \\
&= \frac{\Phi_{\text{LSH}}(U_i, S_i)}{u_i} .
\end{aligned}$$

$\square$

Putting these two lemmas together gives a bound on the expected increase of the potential.

**Lemma E.6 (Lemma 10 in [16])** *For $i \in \{0, 1, \ldots, k-1\}$ and $\mathcal{F}_i$, we have*

$$\mathbf{E}\left[\Psi_{i+1} - \Psi_i \mid \mathcal{F}_i\right] \leq \frac{\Phi_{LSH}(H_i, S_i)}{k-i} .$$

*Proof.* We have that $\mathbf{E}\left[\Psi_{i+1} - \Psi_i \mid \mathcal{F}_i\right]$ equals the sum of

$$\mathbf{E}\left[\Psi_{i+1} - \Psi_i \mid \mathcal{F}_i, \{s \in U_i\}\right] \cdot \Pr[s \in U_i \mid \mathcal{F}_i]$$

and

$$\mathbf{E}\left[\Psi_{i+1} - \Psi_i \mid \mathcal{F}_i, \{s \in H_i\}\right] \cdot \Pr[s \in H_i \mid \mathcal{F}_i] .$$

Now, by Lemma E.4 and Lemma E.5 together with the fact that REJECTIONSAMPLING samples a center in $H_i$ with probability $\frac{\Phi_{\text{LSH}}(H_i, S_i)}{\Phi_{\text{LSH}}(P, S_i)}$, we can upper bound this sum by

$$0 + \frac{\Phi_{\text{LSH}}(U_i, S_i)}{u_i} \cdot \frac{\Phi_{\text{LSH}}(H_i, S_i)}{\Phi_{\text{LSH}}(P, S_i)} \leq \frac{\Phi_{\text{LSH}}(H_i, S_i)}{u_i} \leq \frac{\Phi_{\text{LSH}}(H_i, S_i)}{k-i} .$$

$\square$

We are now ready to bound the overall cost of REJECTIONSAMPLING.

**Theorem E.7 (Theorem 11 in Dasgupta)** *If $S_k = S$ are the centers returned by* REJECTIONSAM-PLING *then*

$$\mathbf{E}\left[\Phi(P, S)\right] \leq 8c^6(\ln(k) + 2) \operatorname{OPT}_k(P) .$$

*Proof.* Using $\Phi(P, S) = \Phi(H_k, S) + \Phi(U_k, S) \leq \Phi(H_k, S) + \Phi_{\text{LSH}}(U_k, S) = \Phi(H_k, S) + \Psi_k$, we have

$$\begin{aligned}
\mathbf{E}\left[\Phi(P, S)\right] &\leq \mathbf{E}\left[\Phi(H_k, S)\right] + \sum_{i=0}^{k-1} \mathbf{E}\left[\Psi_{i+1} - \Psi_i\right] \\
&\leq \mathbf{E}\left[\Phi(H, S)\right] + \sum_{i=0}^{k-1} \mathbf{E}\left[\frac{\Phi_{\text{LSH}}(H_i, S_i)}{k-i}\right] \\
&\leq \mathbf{E}\left[\Phi(H, S)\right] + c^2 \sum_{i=0}^{k-1} \mathbf{E}\left[\frac{\Phi(H_i, S_i)}{k-i}\right] \\
&\leq 8c \cdot \operatorname{OPT}_k(P) + 8c^6 \sum_{i=0}^{k-1} \frac{\operatorname{OPT}_k(P)}{k-i} \\
&\leq 8c^6(\ln(k) + 2) \operatorname{OPT}_k(P) ,
\end{aligned}$$

where the second inequality is by Lemma E.6, the third inequality is by the assumption that the LSH data structure is successful and thus returns $c$-approximate distances, and the penultimate inequality is by Corollary E.3. $\square$

### E.2 Main Theorem for REJECTIONSAMPLING Algorithm

**Theorem 5.4** *For any constant $c > 1$, with probability at least $(1 - 1/n)$ REJECTIONSAMPLING always samples points $x$ that are at most a factor $c^2$ away from the $D^2$-distribution, its expected running time is $O\left(n \log(d\Delta)(d + \log n) + kc^2 d^3 \log(\Delta) \cdot (n \log(\Delta))^{O(1/c^2)}\right)$, and it returns a solution that in expectation is a $O(c^6 \log k)$-approximation of the optimal solution.*

*Proof.* With probability at least $1 - 1/n$, the LSH data structure is successful and we will show that the statements of the theorem holds if that is the case. We start by showing that REJECTIONSAMPLING samples points $x$ that are at most a factor $c^2$ away from the $D^2$-distribution. From Lemma 5.2 we know that the probability of sampling any point $x$ is $\frac{\text{DIST}(x,\text{Query}(x))^2}{\sum_{y \in P} \text{DIST}(y,\text{Query}(y))^2}$. Since (by the assumption that the data structure is successful) we have that $\text{DIST}(x, S) \leq \text{DIST}(x, \text{Query}(x)) \leq c \cdot \text{DIST}(x, S)$, so we have

$$\frac{\text{DIST}(x, S)^2}{c^2 \sum_{y \in P} \text{DIST}(y, S)^2} \leq \frac{\text{DIST}(x, \text{Query}(x))^2}{\sum_{y \in P} \text{DIST}(y, \text{Query}(y))^2}$$

$$\leq c^2 \cdot \frac{\text{DIST}(x, S)^2}{\sum_{y \in P} \text{DIST}(y, S)^2}.$$

The time to initialize the multi-tree embedding (MULTITREEINIT) is $O(nd \log(d\Delta))$, the time to initialize the data structure used by MULTITREEOPEN and MULTITREESAMPLE is $O(n \log(d\Delta))$, the total running time of MULTITREEOPEN is $O(n \log(d\Delta) \log n)$ (Lemma 4.1) and the running time of each call to MULTITREESAMPLE is $O(\log n)$ (Lemma 4.2). Finally, by Lemma 5.3, the expected number of iterations of the loop in REJECTIONSAMPLING is $O(c^2 d^2 k)$, and the running time of each iteration is dominated by the running time of the Insert and Query operations, which is $O\left(d \log(\Delta) \cdot (n \log(\Delta))^{O(1/c^2)}\right)$ by Theorem 5.1. Hence the total running time is $O\left(n \log(d\Delta)(d + \log n) + kc^2 d^3 \log(\Delta) \cdot (n \log(\Delta))^{O(1/c^2)}\right)$. The analysis of the approximation guarantee is presented in the previous section. $\square$

## F  Variance of the Experiments and Aspect Ratio

Table 7 and Table 8 presents the variance of the experiments. Recall that the numbers are reported over 5 runs.

The assumption of bounded aspect ratio allows a clean presentation of the result. The dependency can, for example, be removed (using ideas from prior works) if we have a very rough estimate of the optimum solution (e.g., within a factor $n$ or even $n^{10}$). Indeed, in that case, we can obtain an instance in which each coordinate of each point is an integer in range $[1, \text{poly}(n)]$ by losing a factor $1 + 1/n$ in the approximation guarantee (see [2]). This bounds $\log \Delta = O(\log(nd))$. In practice this can be achieved very efficiently. In order to bound the height of the tree, we propose the following:

- We first compute an estimate of optimum solution by sampling a solution of 20 randomly chosen points from the input. Then we compute the cost of this solution by assigning each point to the closest in the solution.

- Then we divide this value by number of point and number of coordinate and 200. This is intuitively the error that we let each coordinate make. The factor 200 is chosen to ensure that the total error made is within $0.5\%$ of the considered optimum value. The call this value the scaling factor.

- Afterwards, for each dimension of each point we divide it by the scaling factor and remove the fraction. For instance if the value of a considered coordinate is $1.2345$ and scaling factor is $0.01$, the resulting value would be $123$.

| Algorithm | $k = 100$ | $k = 500$ | $k = 1000$ | $k = 2000$ | $k = 3000$ | $k = 5000$ |
|---|---|---|---|---|---|---|
| FASTK-MEANS++ | 75364 | 169739 | 88843 | 92564 | 24225 | 40731 |
| REJECTIONSAMPLING | 288718 | 215658 | 74654 | 68922 | 87984 | 75364 |
| K-MEANS++ | 223686 | 64796 | 26784 | 20958 | 20881 | 30295 |
| AFKMC2 | 393782 | 121318 | 82700 | 22299 | 26945 | 15460 |
| UNIFORMSAMPLING | 687634 | 294580 | 147379 | 189350 | 182828 | 132779 |

Table 7: The variance of the solutions of the algorithms for the Song dataset for various values of $k$. All the numbers are scaled down by a factor $10^5$.

| Algorithm | $k = 100$ | $k = 500$ | $k = 1000$ | $k = 2000$ | $k = 3000$ | $k = 5000$ |
|---|---|---|---|---|---|---|
| FASTK-MEANS++ | 27110 | 672 | 813 | 86 | 77 | 163 |
| REJECTIONSAMPLING | 20440 | 1631 | 799 | 290 | 227 | 86 |
| K-MEANS++ | 8294 | 996 | 269 | 205 | 42 | 24 |
| AFKMC2 | 11529 | 830 | 883 | 204 | 495 | 135 |
| UNIFORMSAMPLING | 567214 | 290954 | 24118 | 23299 | 8770 | 23243 |

Table 8: The variance of the solutions of the algorithm for KDD-Cup dataset for various values of $k$. All the numbers are scaled down by a facto $10^2$.