[Reviews · NeurIPS 2020]

Review 1

Summary and Contributions: The paper shows how to speed up the k-means++ algorithms, without substantial loss in quality of the solution. In particular, the authors consider the case of very high k (think \sqrt{n}), where even one iteration takes O(ndk) time. They reduce the runtime to about O~(nd+n^{1+epsilon}). The quality of the solution, theoretically, drops from O(log k) to O(log k)*(1/epsilon)^{O(1)}. Empirically, the paper shows that, with almost no loss in accuracy, the algorithm is faster, by about an order of magnitude for k's more than 1000 (in datasets of 1/4 to 1/2 million points).

Strengths: - this work speeds-up a very popular algorithm, of wide interest and applicability, - at first it's almost surprising that one can improve on O(ndk): after all just direct computation of which points goes to which cluster would take that time. - the use of hierarchical tree is interesting - emprical results are convincing

Weaknesses: No significant weaknesses. One of course may quibble that there are a number of nuissance 1/epsilon and log factors (some hidden in the O~ notation).

Correctness: yes

Clarity: yes

Relation to Prior Work: yes

Reproducibility: Yes

Additional Feedback:


Review 2

Summary and Contributions: After reading the authors' response, my evaluation remains the same. The submission presents a new and fast algorithm for k-means++ seeding: the authors analysed the approximation guarantee and the runtime fo the presented algorithm; extensive experimental results were also reported.

Strengths: The runtime of the presented algorithm is significantly faster than the previously best-known algorithm for a large value of k; the paper is very well written, and high-level ideas behind the design and analysis of the algorithm is very clear to me; the problem studied in the submission is very relevant to the NeurIPS community.

Weaknesses: The major weakness of the work is the significance of the current work comparing with the previous ones. Specifically, although the runtime of the presented algorithm is faster than the previous algorithms, from their experiments the cost value of the k clusters returned by their new algorithm is significantly worse (about 10%) when k<=1000. I note that the authors did highlight in Footnote 3 the applications in which k-means for a large value of k is needed. However, I am not connived that there're "many practical applications" that require a k-means for k>1000. The lack of applications and the algorithm's weaker performance for k<1000 are the submission's main weakness.

Correctness: The claims and methods are appropriately discussed, and experimental setup is very clear to me.

Clarity: The paper is very well written, and my detailed comments are listed in the Additional feedback session. There's one point that concerns me, i.e. the use of "nearly linear time". Since the algorithm runtime is at least n^{1+epsilon}, this family of algorithms is called *almost linear time algorithms * instead of linear-linear time algorithms. That is, after ignoring poly-log factors, the runtime's dependency of the algorithm is still superlinear in n.

Relation to Prior Work: Relation to prior work is clearly discussed, and I understand how the authors are able to achieve an improvement over the previous algorithms.

Reproducibility: Yes

Additional Feedback: - The algorithm's runtime is function of logDelta, where Delta is the ratio between the maximum distance and minimum distance between two points in the dataset. Now assume that we add an additional point p to the input, and p is arbitrary close to some input point. While it doesn't influence the k-means output for most input instances, introducing p would dramatically increase the value of Delta, and the algorithm's runtime (which doesn't look reasonable to me). Hence, I'm wondering if this log(Delta) term is really needed, or is there some way to redefine Delta, avoid this case, and consequently improve the algorithm's runtime. - Line 186, drop comma before "equals the sum of weights" - Line 214: *from* the D2-distribution - Line 243: We start *by*


Review 3

Summary and Contributions: The goal of the k-means problem is to partition the data to k clusters with minimum cost. k-menas++ is an algorithm for solving the k-means problem which has both theoretical guarantees and used in practice. The first step in the k-means algorithm is to find a good initialization using samples from D^2-distribution. The paper main contribution is to find an approximated way to sample from the D^2-distribution that has theoretical guarantees on the (1) quality of the result, which is comparable to k-means++ guarantees and (2) running time. Importantly, there is a regime where the paper achieves a better running time than other algorithms.

Strengths: The paper designs a new algorithm that is provably better than known algorithms in some regimes. These regimes are suited to several datasets.

Weaknesses: Theory: The proposed algorithm has better running time compared to other methods only if the following two conditions hold (1) k is very large; k >> d (2) aspect ratio is small Experimental: - The new algorithms have running time much better than k-means++ only for large k, but for large k the clusters are very small. For example, in KDD-Cup with k=5000, which showed the greatest improvement in running time, on average, cluster size is only 62. - The authors design two algorithms: "Fast K-means++" and a faster algorithm "Rejection Sampling". Empirically, we see that the theoretically slower algorithm (Fast K-means++) is actually faster.

Correctness: The suggested algorithms and experiments seems sound

Clarity: Yes, the paper is well written. It clearly explains how it compared to other methods, the new algorithms, and the experiments.

Relation to Prior Work: Yes.

Reproducibility: Yes

Additional Feedback:


Review 4

Summary and Contributions: The authors present a near real-time algorithm for k-means++ seeding. The main contribution is a O(nd + (nlog(\Delta)^{1+\eps}) time algorithm that returns a O(log(k) / eps^3) approximate solution. The proposed algorithm is based on three key ingredients; (a) Bartal/FRT's tree embedding to approximate distances from a point to a subset of points (using only 3 trees; Lemma 3.1) (b) a tree data-structure that allows to sample points from kmeans++ D^2 distribution (Corollary 4.3) and (c) rejection sampling along with LSH approximate nearest neighbor data structure (Theorem 5.1). The high level idea is to use a constant number of tree metrics and an additional "sample tree" to maintain an approximate D^2 distribution implicitly. Rejection sampling and LSH is then applied to efficiently sample from the tree data structures.

Strengths: * Solid theoretical results that are well presented * Novel use of rejection sampling and tree embeddings * Speeding up k-means++ is of great importance to practitioners

Weaknesses: * Time complexity-bounds depends on input dataset, i.e., \Delta, i.e., maximum over minimum distane which can be unbounded * Experimental evaluation is based on only 2 datasets. * Resulting algorithm is quite evolved (tree embeddings, LSH and sample tree) to be practical.

Correctness: To the best of my knowledge, all results are stated rigorously and well-explained. I have only a single comment that the authors should explicit mention the probabilistic nature of the main result due to LSH. I will share elaborate on this on my comments below.

Clarity: The paper is well-written and the authors made an effort to clearly present their contribution

Relation to Prior Work: The authors do a great job discussion prior contributions. A notable example if footnote 6.

Reproducibility: Yes

Additional Feedback: Overall: Why only 3 trees are sufficient for Lemma 3.1? Three looks like a magic number after reading the paper. L90-92 you explain the known results that a single tree metric does not suffice, but why three trees? What are the space requirements of the proposed algorithm? L36-41: In your main contribution, you should *not* Use \tilde{O} without defining explicitly the hidden terms. Dependency on \epsilon in the approximation should be explicit. Is it O(logk / eps^3) based on Corollary 5.5. Please be precise. The resulting algorithm is randomized due to the failure probability of LSH. Please transfer this statement to the main result, i.e., having a failure probability. Section 3: MultiTree* methods can be renamed to ThreeTree* since you exactly have three trees. Section 4: You should define earlier what you mean by "open centers". It is not clear until the reader arrives at L136-137. Lemma 4.1: Shouldn't the time complexity depend on k? The expression "opening any set S of k points" is not clear. Maybe you want to state that Algorithm 1 runs in the mentioned time complexity. Lemma 4.2: "is output" -> is sampled (suggestion to rephrase, feel free to ignore) Why Corollary 4.3 is true? Lemma 4.2 returns sampling proabilities that are approximate with respect to D^2 distribution. L198-206 are correct in terms of time complexity but I cannot follow the approximation guarantee. Section 5 L235-236: "We therefore assume throughout the analysis that our data structure is successful". Please remove this assumption when you present the main contribution, i.e, make your statements probabilistic. L277: "O(logn) our algorithm" -> I think the word "dimensions" is missing. Corollary 5.5: make \eps explicit in the bound. Why \Theta here? Can you show a lower bound on the time complexity? Corollary 5.5: "there is an algorithm.." -> you can make this statement more explicity, i.e., there is a randomized algorithm and use failure probability in the statement. ======= After reading the authors feedback, I still consider the submission to be slightly below the acceptance threshold but I lower my confidence level. I still believe that the dependency on \Delta is the weakest point on the submission.

[Author Response · NeurIPS 2020]

We thank the reviewers for their thorough reading of the paper and many insightful and useful comments. Below we outline how we will address the reviewer's comments.

There were two main concerns raised by the reviewers.

1- Running time dependency on the aspect ratio $\Delta$ (max distance over minimum distance).

We believe that in most practical settings the logarithm of the aspect ratio is rather small. The assumption of bounded aspect ratio allows a clean presentation of the result. The dependency can, for example, be removed (using ideas from prior works) if we have a very rough estimate of the optimum solution (e.g., within a factor $n$ or even $n^{10}$). Indeed, in that case, we can obtain an instance in which each coordinate of each point is an integer in range $[1, \mathrm{poly}(n)]$ by losing a factor $1 + 1/n$ in the approximation guarantee, see [1]. This bounds $\log \Delta = O(\log nd)$. In practice this can be achieved very efficiently. We have provided explanations about this in Appendix (line 749-761).

2- Improvement is only for sufficiently large value of k.

We believe that the case of many centers is of similar importance as the case of small k. For example, an important application of k-means is vector quantization, where we need large k to quantize large datasets. See, for example [Cartesian K-Means, Norouzi, Fleet, CVPR'13] for some further discussion.

3- Empirical Evaluation on more datasets.

We will provide experimental evaluation on more data sets. We have run all the algorithms on Census dataset ($n = 2,458,285; d = 29$) as well. The results are very similar to Song dataset. The quality of the solution is comparable with the baselines ($2 - 3\%$ worse than $k$-means++ and almost the same as Afkmc2). Our algorithm is noticeably faster from $k = 1000$ and is $1 - 2$ order of magnitude faster than the baselines for large values of $k$. We will add this to the camera ready version.

**Reviewer 2:** Regarding the quality decrease: We only saw a larger decrease in the quality of the solution for one dataset out of three (and for small $k$).

Regarding the large value of k, please refer to the previous discussion.

We will address the near-linear time and the typos in the camera ready version (thanks for the comments).

**Reviewer 3:** It seems there is a misunderstanding here. FASTkmeans++ is theoretically **faster** than RejectionSampling but it does not come with a theoretical approximation guarantee. In the experiments, it sometimes turns our that Rejection sampling algorithm is slightly faster than Fast k-means++ due the random nature of the algorithm.

For Census dataset, the average cluster size for $k = 5000$ is around 500 and our algorithm is 1-2 orders of magnitude faster. Additionally, we also refer to our discussion of cluster sizes/number of centers above.

**Reviewer 4:** We disagree with the statement that the algorithms are quite involved. We agree that their theoretical analysis is complex, but the implementation is rather simple. We have also submitted the code and will add the code to github after the paper is accepted. The running time of lemma 4.1 does not depend on $k$. The total opening time for all the centers is what mentioned there. We will add Algorithm 1 in this lemma.

Memory requirement is $O(nd + n \log n + n \log \Delta)$, we will add that.

We will make Corollary 5.5 more precise.

Corollary 4.3 does not provide any approximation guarantee and the running time follows from the description of the algorithm. Notice that Lemma 4.2 only states the probability of sampling a point and this does not result in any approximation guarantee for Fast k-means++ algorithm. Indeed it is not clear if the presented Fast k-means++ algorithm has any approximation guarantee, only Rejection sampling algorithm has. We remark that after embedding the point into multiple trees, we do not have the triangle inequality, therefore one cannot simply use the arguments of the proof of the noisy k-means++ algorithm from previous work here to prove an approximation guarantee.

We will also discuss the number of trees selected in more detail. The selection becomes clearer in the proof of Lemma 3.1 in the Appendix but we will add intuition in the camera ready version for this choice.

We will address the remaining editorial comments in the camera ready version.

[1] Better Guarantees for k-Means and Euclidean k-Median by Primal-Dual Algorithms. Sara Ahmadian, Ashkan Norouzi-Fard, Ola Svensson, Justin Ward.

[Meta-Review · NeurIPS 2020]

The paper presents a new algorithm for speeding up k-means++ algorithms with rigorous theoretical guarantees. It is quite surprising that they can improve the running time to \tilde{O}(nd+n^{1+\eps}) when even one round of k-means++ algorithm takes O(ndk) time. The main shortcoming is the performance gain is only visible for large k. However, I think the large k regime is very interesting and does appear in practice. The authors should add discussion about aspect ratio and the new experiments as pointed out by them in the rebuttal.